# Fully computational design of PAM-relaxed *Staphylococcus aureus* Cas9 with expanded targeting capability using UniDesign

Youcai Xiong[1], Li-Kuang Tsai[1], Jun Zhou[1], Shuang Chen[1], Xiaofeng Xia[2], Jifeng Zhang[1], Y Eugene Chen[1], Jie Xu[1]*, Xiaoqiang Huang[1]*

[1]Center for Advanced Models for Translational Sciences and Therapeutics, Department of Internal Medicine, University of Michigan Medical School, Ann Arbor, United States; [2]Research & Development, ATGC Inc, King of Prussia, United States

*For correspondence:
jiex@umich.edu (JX);
xiaoqiah@umich.edu (XH)

## eLife Assessment

This **important** study demonstrates the power of the UniDesign computational framework in prospectively engineering a PAM-relaxed *Staphylococcus aureus* Cas9 variant with editing performance comparable to evolution-derived counterparts. The authors responded promptly and thoroughly to reviewer concerns and strengthened the manuscript with additional experimental validation, providing **compelling** evidence through expanded biochemical characterization across multiple human cell types, comprehensive deep-sequencing analyses, and direct comparisons with established variants that illuminate the mechanistic basis of PAM specificity remodeling and Cas9 optimization. By establishing computational design as a rigorous and viable alternative to directed evolution for CRISPR systems, this work will be of broad interest to the protein engineering, genome engineering, synthetic biology, and computational protein design communities.

**Abstract** CRISPR–Cas9 nucleases have transformed genome engineering, yet their application is often constrained by protospacer-adjacent motif (PAM) requirements. *Staphylococcus aureus* Cas9 (SaCas9) is particularly attractive for in vivo applications due to its compact size; however, its NNGRRT PAM limits targetable genomic sites. Here, we report KRH, a SaCas9 variant designed entirely from the wild-type enzyme through a fully computational point-mutation design workflow, UniDesign, without additional experimental optimization. As expected, KRH efficiently recognizes an expanded NNNRRT PAM and exhibits substantially enhanced editing efficiency at non-canonical PAM sites, with improvements of up to 116-fold over the wild type. KRH achieves genome- and base-editing efficiencies comparable to, or exceeding, those of the well-known evolution-derived KKH variant. Computational modeling by UniDesign provides a mechanistic explanation for the PAM relaxation observed in both KRH and KKH, with structural and energetic analyses revealing that KRH relaxes PAM specificity by fine-tuning the balance between sequence-specific interactions with PAM bases and nonspecific contacts with the DNA backbone. Beyond its practical utility, KRH demonstrates that computational design can identify a minimal set of mutations sufficient to remodel the PAM interface while preserving high nuclease activity. This approach recapitulates—and in some cases surpasses—the performance of evolution-derived variants, offering a scalable strategy for high-throughput Cas9 engineering. Overall, these results establish KRH as a blueprint for rationally engineered, PAM-relaxed nucleases and underscore the power of computational design to accelerate next-generation genome editing.

## Introduction

Clustered regularly interspaced short palindromic repeats (CRISPR) and their associated nucleases have revolutionized biological research and therapeutic genome engineering (*Doudna and Charpentier, 2014*; *Hsu et al., 2014*; *Komor et al., 2017*; *Huang et al., 2022*). Among these systems, *Staphylococcus aureus* Cas9 (SaCas9) is particularly attractive due to its compact size, which facilitates efficient packaging into adeno-associated virus (AAV) vectors for in vivo delivery (*Ran et al., 2015*). However, the utility of SaCas9 is constrained by its relatively restrictive protospacer-adjacent motif (PAM) requirement, NNGRRT, which limits the number of targetable genomic loci (*Nishimasu et al., 2015*). Expanding SaCas9 PAM compatibility while retaining high editing efficiency remains a key challenge for both therapeutic and biotechnological applications.

Previous efforts to relax SaCas9 PAM specificity have relied on molecular evolution, evolution-based chimera engineering, or combined computational–experimental strategies (*Kleinstiver et al., 2015*; *Luan et al., 2019*; *Ma et al., 2019*). The well-known E782K/N968K/R1015H (KKH) variant, generated through iterative mutagenesis and selection, broadens PAM recognition to NNNRRT (*Kleinstiver et al., 2015*). Despite its effectiveness, this approach does not leverage structural information and therefore provides limited mechanistic insight into PAM relaxation. Similarly, chimera engineering methods successfully produced chimeric SaCas9 (cCas9) variants with altered recognition of noncanonical PAM sequences such as NNVRRN and NNVACT (*Ma et al., 2019*), but offered little understanding of the underlying molecular determinants. A combined computational–experimental approach (COMET) explained the activity of KKH and proposed new variants targeting NNGRRN (*Luan et al., 2019*); however, its reliance on intensive molecular dynamics simulations and free-energy perturbation calculations makes it computationally expensive and impractical for high-throughput variant screening.

In our previous work, we demonstrated that UniDesign, a general computational protein design framework, can efficiently and accurately model PAM recognition across diverse Cas9 and Cas12 enzymes (*Huang et al., 2023b*). When native PAM-interacting amino acids (PIAAs) were preserved, UniDesign recovered predicted natural PAMs; conversely, when natural PAMs were fixed, UniDesign redesigned PIAAs recapitulated native PIAA residues with 86% sequence similarity (*Huang et al., 2023b*). These results indicated that UniDesign can capture the intrinsic coupling between PAMs and their interacting residues, suggesting its potential for rationally engineering SaCas9 variants with expanded PAM compatibility.

Here, we report KRH (E782K/N968R/R1015H), a new SaCas9 variant designed entirely in silico using an improved UniDesign workflow for efficient point-mutation generation, without any additional wet-lab optimization. Genome editing and base editing experiments show that KRH robustly recognizes the expanded NNNRRT PAM, matching the relaxed specificity of KKH. Notably, KRH exhibits comparable or superior editing activity across multiple genomic contexts, underscoring its practical utility as a fully computationally designed alternative to evolution-derived variants.

Because KRH was generated exclusively through computational design, our approach also enables detailed interrogation of the structural and energetic mechanisms underlying PAM recognition. Modeling analyses reveal (1) how the KRH substitutions enhance PAM compatibility, (2) how both KRH and KKH achieve PAM relaxation through related yet distinct molecular interactions, and (3) a minimal set of mutations sufficient to remodel the PAM interface without substantially compromising nuclease function. These insights highlight the advantages of UniDesign for rationally CRISPR nuclease engineering.

Together, this work establishes KRH SaCas9 as a powerful, computationally designed nuclease with broadened PAM recognition and demonstrates how atomic-level modeling can both guide and mechanistically explain precise modifications to the PAM-interacting interface. Our results underscore the growing potential of computational protein design approaches like UniDesign to complement traditional molecular evolution strategies for next-generation genome editor engineering.

## Results

### Improving UniDesign for computational point-mutation variant design

UniDesign is a general framework for computational protein design, and we have demonstrated its effectiveness across a wide range of design and modeling tasks (*Huang et al., 2023a*; *Huang et al.,*

*2023b*). In its standard workflow, UniDesign builds rotamers at selected design sites on the input protein scaffold, scores combinations of rotamers using the UniEF energy function (*Huang et al., 2023a*; *Huang et al., 2023b*), and identifies low-energy sequences using simulated annealing Monte Carlo (SAMC) search (*Huang et al., 2020*; *Figure 1A*, light blue). Low-energy sequences from independent trajectories are then collected for downstream analysis.

Although the standard pipeline works well for de novo sequence design and applications that do not restrict the number of mutations—for example, exploring entirely new amino-acid combinations across many positions—it is less suitable for designing variants with only a small, desired number of point mutations, which is often preferred for practical experimental reasons. As a result, in earlier work with UniDesign we had to manually adjust mutable sites one at a time to perform in silico site-directed saturation mutagenesis of CYP102A1 to identify stereoselective point mutations and then combined them (*Huang et al., 2023a*; *Sun et al., 2026*).

Another challenge is redundancy: identical low-energy designs frequently recur across independent SAMC trajectories, reducing design diversity. For example, a particularly favorable single mutant may be sampled repeatedly, sometimes dozens of times, crowding out other potentially informative variants and limiting exploration of the local sequence landscape.

To address these issues, we improve UniDesign's mutant-generation strategy by constraining the number of mutations allowed during SAMC simulations (*Figure 1A*, dark blue). Designs with mutation counts outside the desired range receive a large penalty, biasing the search toward sequences with the specified number of mutations (see 'Methods'). Although this ensures that designs have the intended mutation count, we found that UniDesign still often produced the same low-energy variants multiple times. To reduce this redundancy, we introduced an additional constraint that penalizes any design whose sequence is identical to one already reserved, ensuring that each low-energy design produced by a SAMC trajectory is unique (see 'Methods'). With these two improvements, the enhanced UniDesign pipeline can reliably generate a variety of point-mutation variants (e.g., single, double, or triple mutants) in each run. Nevertheless, because SAMC remains stochastic, multiple runs are still required to obtain sufficient variants and assess design-space convergence.

## Iterative computational design of SaCas9 variants to relax PAM requirements

In our previous work, we showed that UniDesign-derived binding energies correlate strongly with PAM–PIAA interactions: wild-type (WT) PIAAs preferentially recognize canonical PAMs, and native PAMs favor WT PIAAs (*Huang et al., 2023b*). Building on this rationale, we used the enhanced UniDesign pipeline to iteratively design SaCas9 variants with relaxed PAM specificity. WT SaCas9 recognizes the NNGRRT PAM; here, our objective was to expand this specificity to NNNRRT. Using the SaCas9–sgRNA–DNA complex structure (Protein Data Bank (PDB): 5AXW; PAM: TTGGGT) (*Nishimasu et al., 2015*) as the design scaffold, we considered four representative PAMs for PAM-relaxation design: TTAGGT, TTCGGT, TTGGGT, and TTTGGT.

### First design iteration: Reducing positional bias at PAM position 3

We began by targeting Arg1015, which specifically recognizes the third guanine of NNGRRT (*Nishimasu et al., 2015*) via bidentate hydrogen-bonding interactions (*Figure 1B*). For the TTGGGT PAM, WT SaCas9 achieved a binding energy of −514.65 UniDesign energy units (UEU) for TTGGGT, markedly lower than for the other three PAMs (*Supplementary file 1*), confirming the central role of Arg1015 in this positional preference.

Because binding energy is a major determinant of PAM recognition (*Huang et al., 2023b*), we hypothesized that mutations producing similar binding energies across all four PAMs would promote relaxed PAM specificity. To quantify this similarity, we calculated the mean absolute deviation (MAD):

$$\text{MAD} = \frac{1}{n} \sum_{i=1}^{n} \left| e_{\text{bind}}^{i} - \mu \right|, \quad \mu = \frac{1}{n} \sum_{i=1}^{n} e_{\text{bind}}^{i}$$

where $e_{\text{bind}}^{i}$ is the binding energy of a SaCas9 variant for the $i$ th PAM and μ is the mean binding energy for all $n$ PAMs. A MAD of zero indicates identical binding energies.

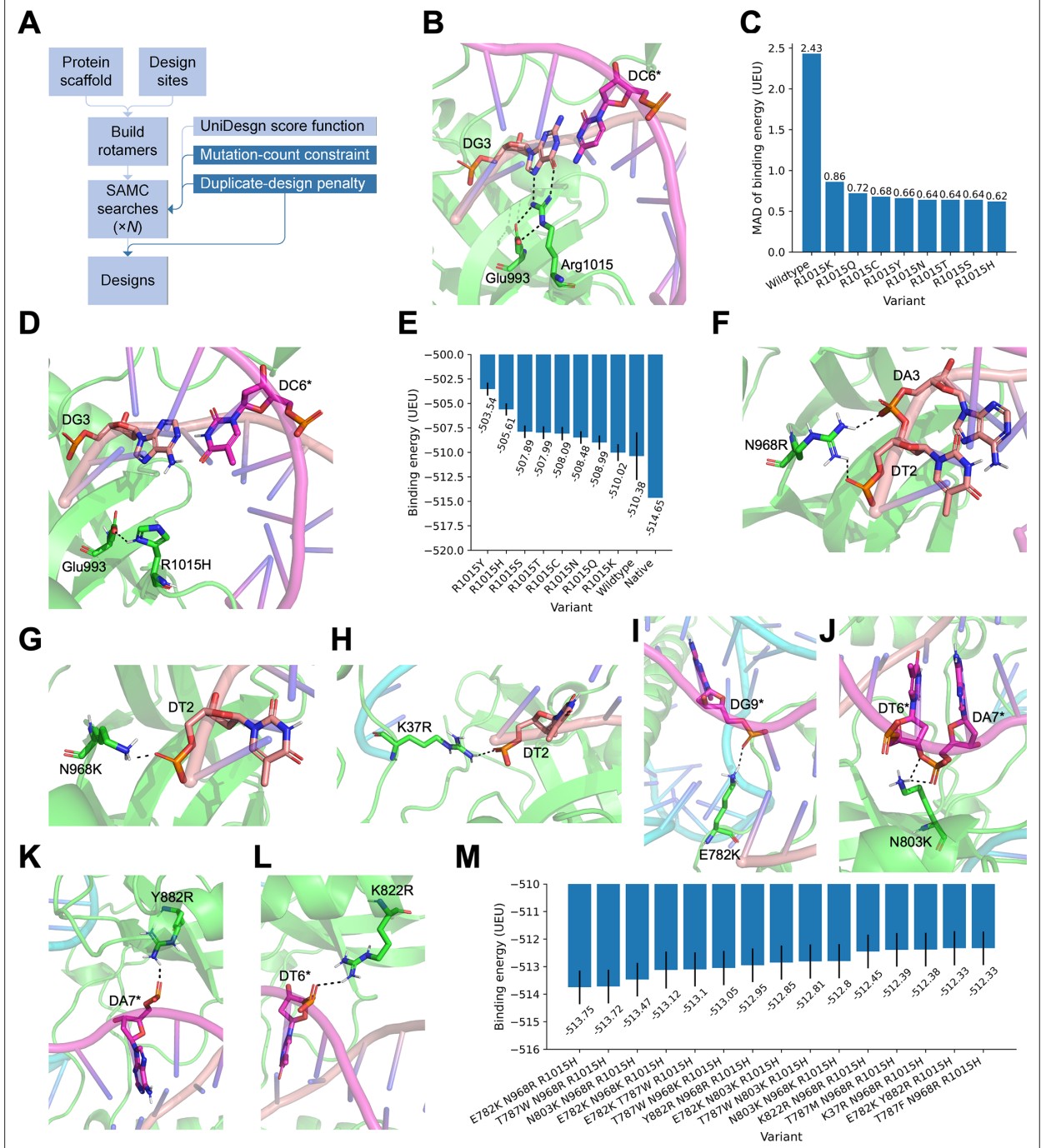

**Figure 1.** Computational design of PAM-relaxed *Staphylococcus aureus* Cas9 variants. (**A**) Improved UniDesign workflow for point-mutation generation. (**B**) Specific recognition of the third guanine in the NNGRRT PAM by Arg1015, positioned through salt-bridge interaction with Glu993. Non-target strand (NTS) nucleotides are indicated by asterisks (as in subsequent panels). (**C**) Mean absolute deviation (MAD) of binding energies across four PAMs (TTAGGT, TTCGGT, TTGGGT, and TTTGGT) for substitutions of Arg1015 with polar or positively charged residues. (**D**) UniDesign model of the R1015H mutant, showing His1015 forming a hydrogen bond with Glu993. (**E**) Mean binding energy across the four PAMs for substitutions of Arg1015 with polar or positively charged residues; error bars represent MADs. (**F–L**) UniDesign models of mutations: N968R (**F**), N968K (**G**), K37R (**H**), E782K (**I**), N803K (**J**), Y882R (**K**), and K822R (**L**). (**M**) Mean binding energies of the top triple-mutant variants across the four PAMs; error bars represent MADs.

The online version of this article includes the following figure supplement(s) for figure 1:

**Figure supplement 1.** UniDesign models of mutations introducing bulky hydrophobic interactions with the DNA backbone.

**Figure supplement 2.** UniDesign RESFILE contents used for SaCas9 redesign.

WT SaCas9 had a high MAD of 2.43 UEU across the four PAMs (*Figure 1C*). To reduce this bias, we performed a single-mutant scan at position 1015, sampling polar and positively charged amino acids (Cys, His, Lys, Asn, Gln, Arg, Ser, Thr, and Tyr). All substitutions substantially reduced the MAD to 0.62–0.86 UEU (*Figure 1C*). Among these, R1015H emerged as a promising substitution because: (1) it achieved a low MAD of 0.62 UEU, indicating effective reduction of positional bias, and (2) histidine, like arginine, can carry a positive charge near physiological pH and can be stabilized by Glu993 via hydrogen-bonding interaction (*Figure 1D*), helping maintain a similar electrostatic environment to the WT. However, the mean binding energy of R1015H (−505.61 UEU) was approximately 9 UEU weaker than the WT for TTGGGT (−514.65 UEU) (*Figure 1E*), suggesting that R1015H alone may not provide sufficiently strong, generalizable DNA binding across TTNGGT PAMs.

## Second design iteration: Restoring binding via nonspecific interactions

To recover binding strength while avoiding sequence-dependent effects, we next introduced additional mutations aimed at enhancing nonspecific protein–DNA interactions. We generated double mutants consisting of R1015H plus one additional mutation at the remaining mutable sites. Across all four PAMs, UniDesign identified 487 double mutants with mean binding energies ranging from −509.77 to −485.86 UEU (*Supplementary file 1*). Of these, 37 variants showed both a mean binding energy < −507.61 UEU and a MAD <0.82 UEU.

Structural analysis revealed that many low-energy designs (e.g., D786Q/M/I/L/V/N; S908R/K/M/I/W; N885F/Y/M/R; T1019L; E782R/T; N785L/K/R/M) primarily enhanced contacts with DNA or sgRNA bases rather than the DNA backbone. Likewise, the E993S/P/A/G/K/R variants relieved some steric clashes between His1015 and the thymine at the second PAM position by altering His1015 geometry, but simultaneously disrupted the favorable, preorganized His1015–Glu993 hydrogen bond (*Figure 1D*). These patterns indicated that such substitutions might not support generalizable PAM relaxation.

In contrast, another subset of double mutants formed favorable backbone-mediated, DNA sequence-nonspecific interactions. These included mutations such as N968R/K, E782K, T787W/F/M, N803K/W, Y882R, K37R, and K822R. These positively charged arginine or lysine residues were capable of forming salt bridges with DNA backbone phosphate groups (*Figure 1F–L*), while the bulky hydrophobic residues could form favorable packing interactions (*Figure 1—figure supplement 1*). These variants achieved binding energies between −507.76 UEU (N803W/R1015H) and −509.70 UEU (N968R/R1015H) (*Supplementary file 1*), although none matched the WT energy of −514.65 UEU for the native TTGGGT PAM.

## Third design iteration: Combining favorable substitutions

To further strengthen binding, we generated triple mutants by combining promising double-mutant substitutions. Across all four PAMs, UniDesign produced 50 triple mutants, all of which had lower mean binding energies than their corresponding double mutants, indicating largely additive energetic contributions (*Supplementary file 1*). The top-performing designs combined E782K, N968R/K, T787W, N803K, or Y882R with R1015H. The best triple mutant, E782K/N968R/R1015H (denoted KRH), achieved a mean binding energy of −513.75 UEU and a MAD of 0.61 UEU, closely matching the WT energy for TTGGGT (−514.65 UEU) (*Figure 1M*). The previously reported KKH variant (E782K/N968K/R1015H) was also recapitulated and ranked fourth by mean binding energy (−513.12 UEU; MAD 0.67 UEU). Other top-ranked variants included T787W/N968R/R1015H, N803K/N968R/R1015H, and E782K/T787W/R1015H (*Figure 1M*; *Supplementary file 1*).

Because the known KKH variant exhibits strong editing activity across NNGRRT PAMs (*Kleinstiver et al., 2015*)—and given the close structural and energetic similarity between KKH and our top-ranked KRH variant—we anticipated that KRH may display comparable or superior genome-editing performance. We therefore proceeded to experimental assessment without an additional design iteration.

## KRH expands the targeting range of SaCas9 across diverse cell types

To systematically evaluate whether the computationally designed KRH variant can expand the PAM range from NNGRRT to NNNRRT, we compared the editing efficiencies of WT and KRH SaCas9 across multiple genomic targets with distinct PAM sequences.

In HEK293T cells, KRH showed markedly improved editing efficiency compared to the WT at non-canonical PAM sites (NNHRRT, H=A/C/T) (*Figure 2A*). The enhancement ranged from 1.4-fold to 116.3-fold. For example, at the *RUNX1* locus, editing efficiency increased from 0.47% with SaCas9 to 54.68% with KRH, a 116.3-fold improvement. Although efficiency at a classical NNGRRT PAM site (*VEGFA*) modestly decreased from 80.19% (WT) to 67.37% (KRH), overall activity remained high. These results indicate that KRH broadens the PAM compatibility of SaCas9 while maintaining strong activity at canonical PAMs.

To assess whether this PAM-relaxing effect is generalizable across cell types, we next compared KRH and WT SaCas9 in A549, HeLa, and U2OS cells using targets with various PAM sequences (*Figure 2B–D*). In all three cell types, KRH exhibited significantly higher editing efficiencies at non-classical PAM sites while maintaining comparable activity to WT at classical NNGRRT sites. Editing efficiencies varied among cell types, suggesting that SaCas9 activity is influenced by cellular context; nevertheless, KRH consistently outperformed WT at non-classical PAMs (*Figure 2E*). On average, KRH increased editing efficiency by 5.4-fold in HEK293T cells, 4.3-fold in A549 cells, 6.9-fold in HeLa cells, and 8.4-fold in U2OS cells.

We also compared the performance of KRH and WT SaCas9 across different PAM categories (*Figure 2F*). At NNARRT, NNCRRT, and NNTRRT PAMs, KRH improved editing efficiency by an average of 3.3-fold, 10.9-fold, and 15.4-fold, respectively. At canonical NNGRRT PAMs, KRH generally retained activity comparable to WT SaCas9.

Taken together, these results demonstrate that the computationally designed KRH SaCas9 variant substantially broadens the PAM range of WT SaCas9 and enables efficient genome editing at sites with non-classical PAM sequences across multiple human cell types.

## KRH SaCas9-based adenine base editor (ABE) broadens base-editing capabilities

Given that KRH expands the PAM compatibility of SaCas9, we next examined whether a base editor engineered from KRH could target a broader set of genomic regions. To this end, we constructed KRH-ABE by fusing the ABE8e (V106W) deaminase to the N-terminus of KRH nickase (D10A) (*Richter et al., 2020*). For an appropriate comparison, we also generated a WT SaCas9 (D10A)-based ABE, termed WT-ABE. We then systematically compared the editing efficiencies of WT-ABE and KRH-ABE across four cell types (HEK293T, A549, HeLa, and U2OS) at genomic loci bearing diverse PAM sequences.

In HEK293T cells, KRH-ABE exhibited substantially higher base-editing efficiencies than WT-ABE at all tested loci, including those with classical NNGRRT PAMs (*Figure 3A*). At the *EMX1* locus, the A11 editing efficiency increased from 14.5% with WT-ABE to 81.1% with KRH-ABE, representing a 5.6-fold enhancement; at the *RUNX1* locus, the A15 editing efficiency increased from 0.3% with WT-ABE to 19.5% with KRH-ABE, representing a 65-fold enhancement. In A549, HeLa, and U2OS cells, KRH-ABE again showed consistently higher editing efficiency at non-classical PAM sites, while maintaining comparable activity to WT-ABE at classical PAM targets (*Figure 3B–D*). These results demonstrate that KRH-ABE can target a wider set of genomic sites than the WT SaCas9-based editor.

As observed with genome editing, ABE performance varied across cell types (*Figure 3A–D*), but KRH-ABE consistently outperformed WT-ABE at non-classical PAM loci in every cell line tested. On average, KRH-ABE increased editing efficiency by 9.4-fold in HEK293T cells, 3.5-fold in A549 cells, 4.6-fold in HeLa cells, and 6.3-fold in U2OS cells (*Figure 3E*).

We also compared performance across PAM categories (*Figure 3F*). KRH-ABE improved editing efficiency by an average of 6.5-fold at NNARRT PAM sites, 4.7-fold at NNCRRT sites, and 6.5-fold at NNTRRT sites. At canonical NNGRRT PAMs, KRH-ABE displayed slightly higher (but overall comparable) efficiency relative to WT-ABE.

Finally, we assessed potential off-target (OT) effects. KRH-ABE showed OT edits at rates comparable to WT-ABE, indicating that the KRH variant maintains the low OT activity characteristic of WT SaCas9 (*Figure 3—figure supplement 1*).

Collectively, these results show that KRH-ABE substantially broadens the targeting range of SaCas9-based base editors while preserving high specificity, thereby experimentally validating the computational predictions underlying the KRH design.

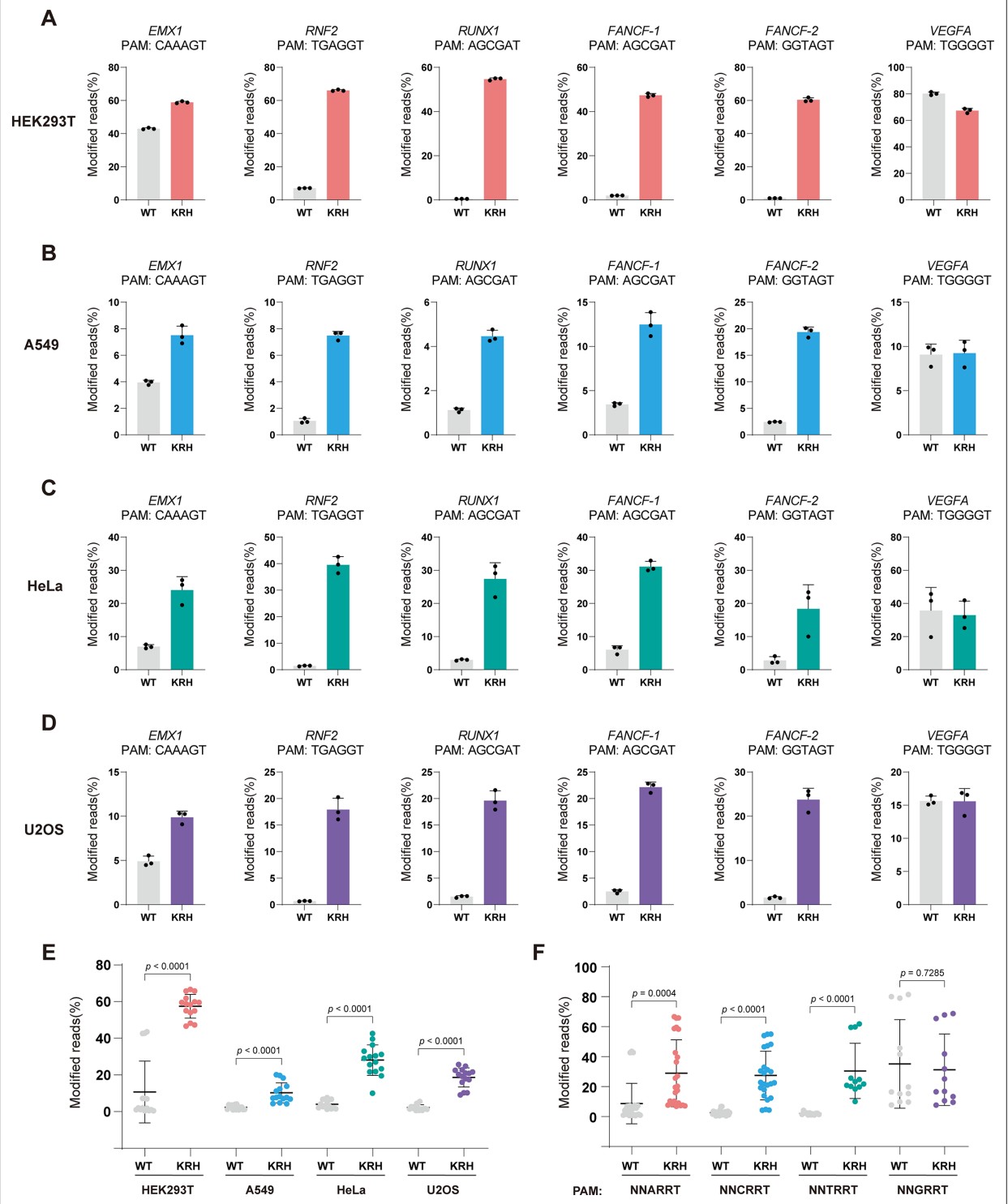

**Figure 2.** KRH expands the targeting range of SaCas9 across diverse cell types. (**A**) Editing efficiencies of wild-type (WT) SaCas9 and the KRH variant at different PAM sites in HEK293T cells. Bars represent the mean of n=3 independent biological replicates; error bars indicate s.d. (**B**) Editing efficiencies of WT SaCas9 and the KRH variant at different PAM sites in A549 cells. Bars represent the mean of n=3 independent biological replicates; error bars indicate s.d. (**C**) Editing efficiencies of WT SaCas9 and the KRH variant at different PAM sites in HeLa cells. Bars represent the mean of n=3 independent biological replicates; error bars indicate s.d. (**D**) Editing efficiencies of WT SaCas9 and the KRH variant at different PAM sites in U2OS cells. Bars represent the mean of n=3 independent biological replicates; error bars indicate s.d. (**E**) Statistical comparison of editing efficiencies between WT SaCas9 and the KRH variant at non-canonical PAM sites across HEK293T, A549, HeLa, and U2OS cells. Data are shown as mean ± s.d.; statistical significance was assessed using a two-tailed unpaired Student's *t*-test. (**F**) Statistical comparison of editing efficiencies between WT SaCas9 and the KRH

*Figure 2 continued on next page*

*Figure 2 continued*

variant across different PAM classes (NNARRT, NNCRRT, NNTRRT, and NNGRRT). Data are shown as mean ± s.d.; statistical significance was assessed using a two-tailed unpaired Student's *t*-test.

## KRH SaCas9 demonstrates editing efficiency comparable to KKH SaCas9

The previously reported KKH variant also relaxes the PAM preference of SaCas9, enabling recognition of the same NNNRRT PAM motif (*Kleinstiver et al., 2015*). Thus, a direct comparison between KKH and our newly designed KRH variant is conducted. Structurally, both triple mutants share two key features: in each, the E782K mutation forms a favorable salt bridge with the backbone phosphate of non-target strand (NTS) nucleotide 9 (e.g., DG9*), and the R1015H mutation is stabilized by Glu993 through an oriented hydrogen bond (*Figure 4A*). The only structural differences arise at residue 968. In KKH, the N968K substitution forms a single salt bridge with the phosphate of target strand (TS) nucleotide 2 (e.g., DT2), whereas in KRH, the N968R substitution forms two salt bridges with the phosphate groups of TS nucleotides 2 and 3 (e.g., DT2 and DA3) (*Figure 4A*).

To directly compare their activities, we evaluated the editing efficiencies of KKH and KRH across multiple genomic loci bearing different PAM sequences. At all tested sites, KRH exhibited editing efficiencies comparable to those of KKH (*Figure 4B*). Overall, both variants displayed similar performance, without systematic advantages for either enzyme (*Figure 4C*).

We also compared base-editing activities of ABE systems constructed from each variant (KKH-ABE vs. KRH-ABE). Consistent with the genome-editing results, the two base editors showed comparable efficiencies across all tested PAM contexts (*Figure 4D and E*).

## Discussion

In this study, we demonstrate that a fully computational protein design strategy based on UniDesign can generate SaCas9 variants with relaxed PAM specificity and high genome-editing activity. The KRH variant matches the performance of the evolution-derived KKH, establishing UniDesign as a predictive framework for PAM engineering rather than a descriptive or post hoc tool. Moreover, this study highlights that UniDesign enables rapid and scalable exploration of sequence space without requiring iterative experimental screening, offering a complementary—and potentially more efficient—alternative to directed evolution for future protein engineering applications.

Mechanistically, our results indicate that PAM recognition and editing activity are governed by a balance between sequence-specific interactions with PAM bases and nonspecific interactions with the DNA backbone. Fine-tuning this balance of specificity and nonspecificity provides a rational strategy for engineering Cas9 variants with relaxed PAM requirements. UniDesign modeling revealed that WT SaCas9 exhibits a strong binding preference for the canonical TTGGGT PAM relative to alternative PAMs (TTAGGT, TTCGGT, and TTTGGT), and that Arg1015 is a key molecular determinant mediating specific recognition of the guanine at the third PAM position. Disrupting this specific interaction by substituting Arg1015 with His substantially reduced SaCas9 activity toward the TTGGGT PAM, whereas the R1015A mutation nearly abolished editing activity altogether (*Kleinstiver et al., 2015*). These effects are consistent with UniDesign predictions, which indicate a marked reduction in binding interactions upon disruption of the Arg1015–guanine hydrogen-bonding interactions (*Figure 1E*; *Supplementary file 1*). Together, these observations indicate that excessively weakened binding to the native PAM is insufficient to support robust editing. Consequently, enhancing nonspecific Cas9–DNA interactions emerges as a natural and effective strategy to restore activity while preserving equivalent recognition of relaxed PAMs at the third PAM position. Through iterative, fully computational design, we identified triple mutants with mean binding energies comparable to that of WT SaCas9 on the native TTGGGT PAM, a prediction that was fully supported by subsequent experimental validation.

Compared with prior approaches, our method is distinguished by its fully computation-driven nature and minimal reliance on empirical screening or expert heuristics. While previous studies achieved effective PAM relaxation, they typically required extensive wet-lab evolution and/or computationally expensive simulations. In contrast, the improved UniDesign workflow enables efficient and direct exploration of promising point mutations. Depending on the number of design sites, variants generated, and stages of iterative design, a single UniDesign run on one PAM model (e.g., TTAGGT)

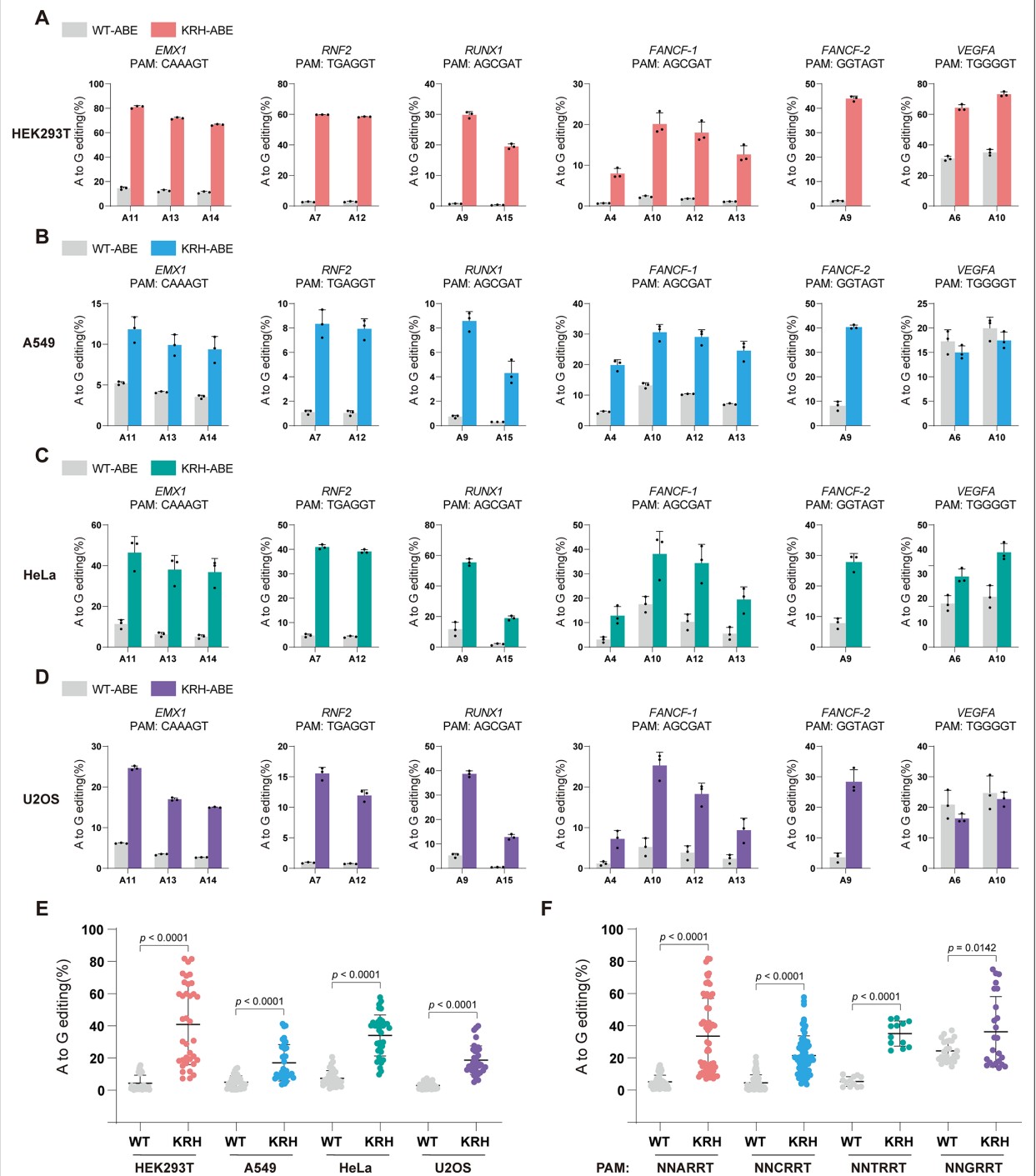

**Figure 3.** KRH-based ABE further broadens base-editing capabilities. (**A**) Base-editing efficiencies of WT-ABE and KRH-ABE at different PAM sites in HEK293T cells. Bars represent the mean of n=3 independent biological replicates; error bars indicate s.d. (**B**) Base-editing efficiencies of WT-ABE and KRH-ABE at different PAM sites in A549 cells. Bars represent the mean of n=3 independent biological replicates; error bars indicate s.d. (**C**) Base-editing efficiencies of WT-ABE and KRH-ABE at different PAM sites in HeLa cells. Bars represent the mean of n=3 independent biological replicates; error bars indicate s.d. (**D**) Base-editing efficiencies of WT-ABE and KRH-ABE at different PAM sites in U2OS cells. Bars represent the mean of n=3 independent biological replicates; error bars indicate s.d. (**E**) Statistical comparison of base-editing efficiencies between WT-ABE and KRH-ABE at non-canonical PAM sites across HEK293T, A549, HeLa, and U2OS cells. Data are shown as mean ± s.d.; statistical significance was assessed using a two-tailed unpaired Student's t-test. (**F**) Statistical comparison of base-editing efficiencies between WT-ABE and KRH-ABE across different PAM classes (NNARRT, NNCRRT, NNTRRT, and NNGRRT). Data are shown as mean ± s.d.; statistical significance was assessed using a two-tailed unpaired Student's t-test.

The online version of this article includes the following figure supplement(s) for figure 3:

**Figure supplement 1.** Evaluation of off-target (OT) effects of WT-ABE and KRH-ABE at predicted OT sites.

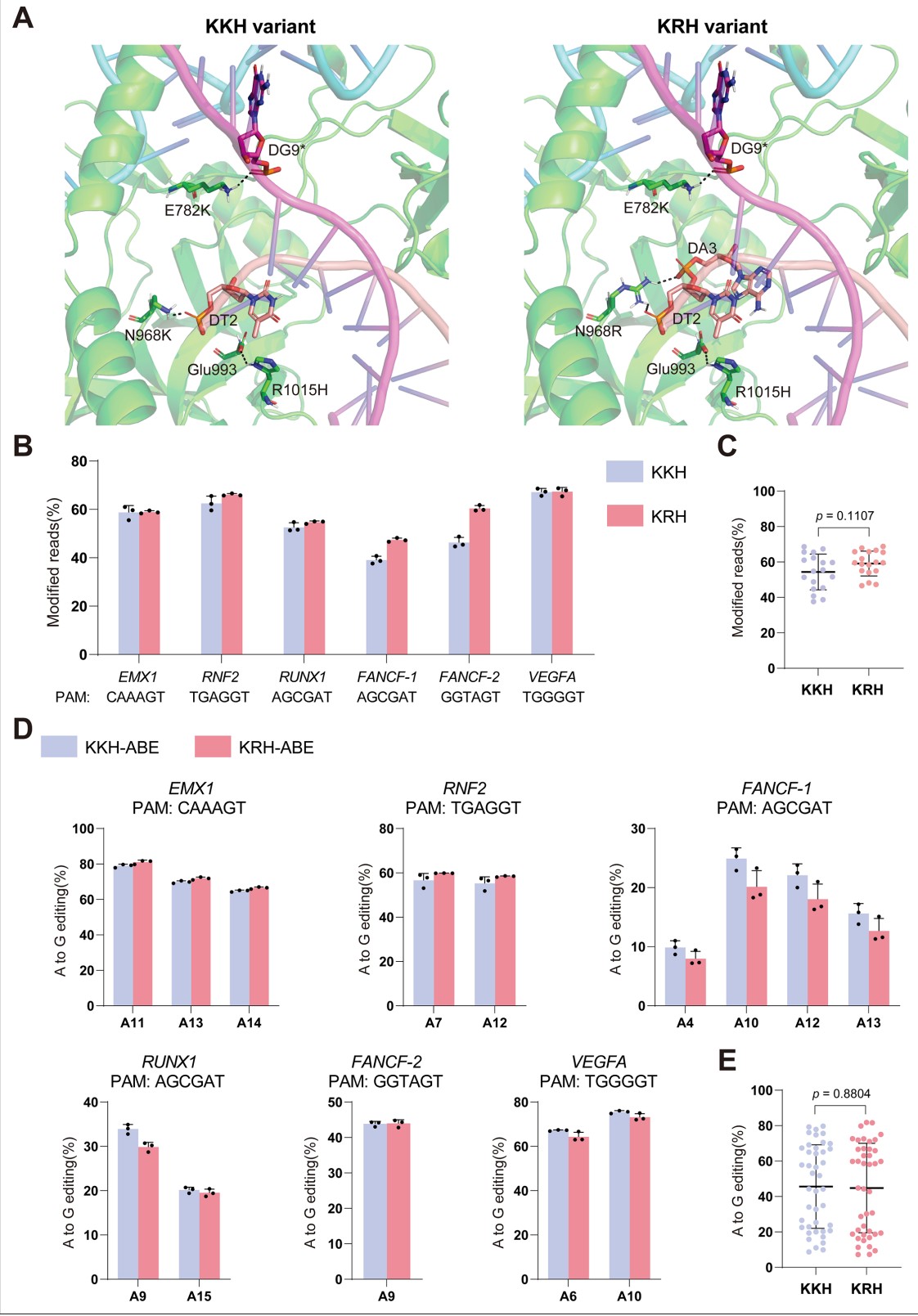

**Figure 4.** Comparison of KKH and KRH *Staphylococcus aureus* Cas9 variants. (**A**) UniDesign models of the KKH and KRH variants. Non-target strand (NTS) nucleotides are indicated by asterisks. (**B**) Editing efficiencies of the KKH and KRH variants at different PAM sites in HEK293T cells. Bars represent the mean of n=3 independent biological replicates; error bars indicate s.d. (**C**) Statistical comparison of editing efficiencies between the KKH and KRH variants in HEK293T cells. Data are shown as mean ± s.d.; statistical significance was assessed using a two-tailed unpaired Student's *t*-test. (**D**) Base-

*Figure 4 continued on next page*

*Figure 4 continued*
editing efficiencies of KKH-ABE and KRH-ABE in HEK293T cells. Bars represent the mean of n=3 independent biological replicates; error bars indicate s.d. (**E**) Statistical comparison of base-editing efficiencies between KKH-ABE and KRH-ABE in HEK293T cells. Data are shown as mean ± s.d.; statistical significance was assessed using a two-tailed unpaired Student's *t*-test.

can be completed in minutes to several hours, making our approach well suited for rapid and scalable Cas9 variant discovery.

While the KRH variant differs from the experimentally evolved KKH by only a single residue, to the best of our knowledge, the N968R mutation has not been experimentally characterized prior to this study and was identified through our computational design workflow. From a practical perspective, although the KKH variant exhibits robust editing activity comparable to that of KRH, we consistently observed higher editing efficiencies for KRH at a subset of target sites (*Figure 4B*). This site-dependent advantage suggests that KRH may achieve a more favorable balance between PAM flexibility and target DNA engagement, potentially resulting in enhanced catalytic efficiency under certain sequence or chromatin contexts. Accordingly, KRH represents a compelling alternative to KKH in applications where maximal editing efficiency is required, particularly for targets that are suboptimal or only weakly accessible to existing SaCas9 variants. More broadly, these results demonstrate that computationally designed variants can not only recapitulate but, in some cases, surpass the performance of evolution-derived nucleases.

Finally, we emphasize that SaCas9 holds significant therapeutic potential for CRISPR-based disease applications due to its compact size, which enables packaging into a single AAV vector. The development of the KRH variant expands both the targeting range and efficiency of SaCas9, representing an important advance that may reduce reliance on dual-AAV split-Cas9 delivery strategies, which are often associated with lower efficiency (*Chew et al., 2016*; *Moreno et al., 2018*).

This study has several limitations. First, our computational design relies on the availability of high-quality Cas9 structures in complex with target DNA. Although high-resolution experimental structures exist for extensively studied enzymes such as SpCas9 (*Anders et al., 2014*; *Jinek et al., 2014*; *Nishimasu et al., 2014*) and SaCas9, they are not always available for other Cas proteins of interest. This limitation may be partially mitigated by advances in deep learning-based structure prediction methods, such as AlphaFold3 (*Abramson et al., 2024*), which can generate high-quality models of protein-DNA complexes. Second, although KRH exhibited high editing efficiency at both canonical and relaxed PAMs in HEK293T cells (approximately 50–70% modified reads; *Figure 2A*), substantially lower activity was observed at the tested sites in other cell types, particularly in A549 cells (approximately 4–20% modified reads; *Figure 2B*). This cell-type dependence likely reflects differences in chromatin accessibility and/or the expression levels of key DNA repair pathway components. Further engineering of SaCas9 to enhance activity in such 'hard-to-edit' cellular contexts will be an important direction for future work.

Looking forward, the UniDesign framework and design strategy presented here can be extended to other Cas enzymes, providing a general approach for engineering CRISPR enzymes toward diverse functional goals. For instance, we are currently assessing the applicability of this strategy to the design of PAM-relaxed Cas8 variants. Moreover, recent work by Shi et al. revealed a two-step target capture mechanism for efficient CRISPR–Cas9 editing, in which highly PAM-relaxed Cas9 variants like SpRY can become kinetically trapped between steps, reducing overall editing efficiency (*Shi et al., 2025*). This suggests that, rather than relying on a single broadly PAM-compatible Cas9, a catalog of Cas9 variants—each optimized for a subset of PAM sequences with high specificity—may represent a more effective strategy for high-efficiency, high-fidelity genome editing (*Collias and Beisel, 2021*). We anticipate that UniDesign and continued methodological improvements will facilitate the rational development of such Cas9 catalogs for next-generation genome editing applications.

## Methods
### SaCas9 structure preprocessing and PAM variant generation
The crystal structure of SaCas9 bound to sgRNA and target DNA (PDB: 5AXW; PAM: TTGGGT) was downloaded from the Protein Data Bank and processed as previously described (*Huang et al., 2023b*).

Specifically, UniDesign's *RepairStructure* and *Minimization* modules were applied to add missing side-chain atoms and optimize side-chain conformations to reduce steric clashes.

To evaluate our computational approach for relaxing PAM specificity, we focused on the most strictly required nucleotide within the SaCas9 PAM—the guanine at the third position of NNGRRT. Using the preprocessed 5AXW structure as a template, we used UniDesign's *BuildMutant* function to generate four structural models differing only in the PAM and its base-paired positions: TTAGGT, TTCGGT, TTGGGT, and TTTGGT. These variants enabled systematic assessment of sequence-dependent structural changes introduced by alternative nucleotides at the critical PAM position.

## Design sites selection

Protein design sites were classified into two categories: mutable and repackable. Mutable sites are residues allowed to change amino-acid identity during design. Repackable sites are neighboring residues whose side-chain conformations may be adjusted to accommodate mutations at mutable sites; these residues are not themselves mutated. Using the 5AXW structure visualized in PyMOL, we selected mutable and repackable sites according to the following rules.

Mutable sites included: (1) Arg1015, which directly recognizes the third guanine of the PAM; (2) residues with any side-chain atom within 4.5 Å of the side chain of Arg1015; and (3) residues with any side-chain atom within 6 Å of the main-chain atoms of the PAM, its reverse-complement nucleotides, and the nucleotides located three bases upstream and downstream of both the PAM and its reverse complement.

Repackable sites were defined as protein residues whose side-chain atoms lie within 4.5 Å of the side-chain atoms of any mutable site, allowing localized structural relaxation during design.

The final mutable sites were Lys37, Glu782, Asn785, Asp786, Thr787, Tyr789, Asn803, Lys815, Lys818, Lys822, Tyr882, Asn885, Asn888, Ala889, Ser908, Leu909, Lys910, Pro911, Asn968, Ile982, Leu988, Glu993, Arg1002, Arg1012, Pro1013, Arg1015, and Thr1019.

The final repackable sites were Ile784, Ile801, Asn804, Leu805, Asn813, Asp814, Leu827, Val905, Leu907, Lys929, Asn930, Arg980, Leu989, Arg991, Asn995, Tyr1001, Asp1010, Pro1014, and Ile1017.

## Improvement of UniDesign for designing point-mutation variants

Relative to earlier UniDesign versions (***Huang et al., 2023a***; ***Huang et al., 2023b***), the main enhancement in the current version (v1.2) is the introduction of new score terms that allow explicit control over the number of mutations in designed variants. Below, we summarize these updates.

The original UniDesign scoring function is defined as

$$E_{\text{total}} = E_{\text{UniEF,non-bind}} + w_{\text{UniEF,bind}} E_{\text{UniEF,bind}} + w_{\text{evo}} E_{\text{evo}}$$

Here, $E_{\text{UniEF,non-bind}}$ and $E_{\text{UniEF,bind}}$ are non-binding and binding energies computed using the UniEF energy function. $E_{\text{evo}}$ is an evolutionary term used primarily for de novo protein sequence design. The weights $w$ control the relative contributions of each term, and by default both $w_{\text{UniEF,bind}}$ and $w_{\text{evo}}$ are set to 1. The evolutionary term is optional and typically disabled in protein redesign tasks.

To bias the design process toward generating variants with a specific number of point mutations, two additional penalty terms were introduced. The updated scoring function is

$$E_{\text{total}} = E_{\text{UniEF,non-bind}} + w_{\text{UniEF,bind}} E_{\text{UniEF,bind}} + w_{\text{evo}} E_{\text{evo}} + w_{\text{OOR}} N_{\text{OOR}} + w_{\text{exist}} X_{\text{exist}}$$

The first three terms match the original score function. The newly added terms are:

### Mutation-count penalty ($w_{\text{OOR}} N_{\text{OOR}}$)

This term penalizes designs whose total number of mutations ($n_{\text{mut}}$) falls outside the user-specified mutation-count range $[N_{\text{mut,min}}, N_{\text{mut,max}}]$:

$$N_{\text{OOR}} = \begin{cases} N_{\text{mut,min}} - n_{\text{mut}}, & \text{if } n_{\text{mut}} < N_{\text{mut,min}} \\ n_{\text{mut}} - N_{\text{mut,max}}, & \text{if } n_{\text{mut}} > N_{\text{mut,max}} \\ 0, & \text{otherwise} \end{cases}$$

This encourages the SAMC search to remain within the desired mutation-count range (e.g., exactly one mutation, or up to two mutations).

## Duplicate-design penalty ($w_{\text{exist}}X_{\text{exist}}$)

To prevent the same low-energy variant from being repeatedly generated across independent SAMC trajectories, designs that match a previously saved sequence are penalized:

$$X_{\text{exist}} = \begin{cases} 1, & \text{if the design has already been saved} \\ 0, & \text{otherwise} \end{cases}$$

This promotes exploration of diverse sequence space.

Both weights $w_{\text{OOR}}$ and $w_{\text{exist}}$ are set to large positive constants (1000 by default and in this study) to strongly enforce adherence to mutation-count constraints and uniqueness.

In implementation, we did not remove or modify the original UniDesign functionalities for de novo sequence design or applications that do not restrict the number of mutations. Instead, we introduced new optional parameters that allow users to control the mutant-design strategy when desired. Specifically, the options `--min_muts`, `--max_muts`, and `--penalize_identical_seqs` enables users to specify the allowable mutant-count range and to penalize the generation of duplicate sequences. When these options are not used, UniDesign behaves identically to previous versions.

## Computational mutant design with improved UniDesign

All computational protein design simulations were performed using UniDesign v1.2. For each design iteration, the input structural model was the preprocessed SaCas9–sgRNA–DNA complex derived from 5AXW, as described above. Design-site constraints were specified using UniDesign's RESFILE format (*Huang et al., 2023a*; *Huang et al., 2023b*), and design simulations were performed using the SAMC protocol with mutation-count constraints and duplicate-design penalties enabled. SAMC trajectories were run using the default UniDesign temperature schedule. Unless otherwise stated, each design iteration generated up to 1000 SAMC trajectories (variants) and was repeated independently 10 times. Among the 10 repeats, the three with the lowest total energy for each PAM model and SaCas9 variant were used for energetic analysis to assess convergence.

### Design iteration 1: Identifying mutations at Arg1015 that relax PAM-position specificity

The first design stage focused on exploring amino-acid substitutions at Arg1015, the residue that directly recognizes the third guanine in the NNGRRT PAM. To limit the search to chemically plausible alternatives and avoid destabilizing mutations, we allowed only polar or positively charged amino-acid types at position 1015: {Cys, His, Lys, Asn, Gln, Ser, Thr, Tyr, and the native Arg}. All other mutable residues were fixed to their native identities. These constraints were encoded in a RESFILE, whose content is shown in *Figure 1—figure supplement 2A*.

To ensure that at most single-mutation variants were produced, the minimum and maximum allowed mutation counts were set to 0 and 1, respectively. Because position 1015 had only nine allowed amino-acid identities, at most nine unique variants were generated, even though the maximum number of trajectories was set to 1000. Candidate substitutions were evaluated based on total energy, binding energy across four PAM models, and structural inspection for steric compatibility. This stage identified R1015H as the most promising mutation for reducing PAM-position bias.

### Design iteration 2: Designing double mutants with R1015H as a fixed background

The second design stage aimed to identify additional mutations that, when combined with R1015H, improved while balancing the mean binding energy across all four modeled PAM variants (TTAGGT, TTCGGT, TTGGGT, and TTTGGT). To accomplish this, residue 1015 was fixed to His, while all other mutable sites were allowed to sample all 20 amino-acid types. The RESFILE content for this iteration is provided in *Figure 1—figure supplement 2B*.

The minimum and maximum allowed mutation counts were both set to 2 in this iteration. Each SAMC trajectory generated one unique low-energy double mutant, enforced by sequence-uniqueness constraints. After pooling results across independent runs, candidate double mutants were ranked based on (1) mean binding energy across all four PAM models, (2) structural plausibility, assessed by eliminating variants with steric clashes or unrealistic side-chain orientations, and (3) consistency, assessed by the recurrence of the same mutation across multiple independent SAMC trajectories. Promising second-site mutations identified in this stage included: N968R/K, E782K, T787W/F/M, N803K/W, Y882R, K37R, and K822R.

## Design iteration 3: Combining promising mutations into higher-order variants

The third design stage sought to combine the beneficial second-site mutations identified in design iteration 2 into higher-order variants. To restrict the combinatorial space while enriching for functional combinations, each promising position was allowed to sample only the favorable amino-acid types identified previously or its native identity. These constraints were encoded in the RESFILE as follows: Lys37 → {Lys, Arg}; Glu782 → {Glu, Lys}; Thr787 → {Thr, Met, Phe, Trp}; Asn803 → {Asn, Lys, Trp}; Lys822 → {Lys, Arg}; Tyr882 → {Tyr, Arg}; and Asn968 → {Asn, Lys, Arg}. All other mutable sites were fixed to their native identities, and residue 1015 was kept as His for all designs. The RESFILE content for this iteration is provided in *Figure 1—figure supplement 2C*.

The minimum and maximum allowed mutation counts were both set to 3 in this iteration. SAMC runs were repeated using the same conditions as above, and variants that consistently emerged across independent runs and exhibited favorable energy profiles were analyzed and selected for experimental characterization.

## Plasmids construction

The KKH SaCas9 plasmid was obtained from Addgene (Plasmid #70708). Based on this plasmid, the WT and KRH variants were generated by GenScript. WT-, KKH-, and KRH-based ABE constructs were also synthesized by GenScript. The sequences of all constructs are provided in *Supplementary file 2*.

Each sgRNA expression cassette was synthesized and subsequently cloned into the pUC-GW-Amp vector (Genewiz). The sequences of all sgRNAs used in this study are listed in *Supplementary file 2*.

## Cell culture, transfection, and genomic DNA isolation

HEK293T (ATCC, CRL-11268) and A549 (ATCC, CCL-185) cells were cultured in Dulbecco's Modified Eagle Medium (Gibco) supplemented with 10% FBS (Gibco) and 1% penicillin-streptomycin (Gibco). HeLa (ATCC, CCL-2) cells were maintained in Eagle's Minimum Essential Medium (ATCC) supplemented with 10% FBS (Gibco) and 1% penicillin-streptomycin (Gibco). U2OS (ATCC, HTB-96) cells were maintained in McCoy's 5 A Medium (ATCC) supplemented with 10% FBS (Gibco) and 1% penicillin-streptomycin (Gibco). Cell line identity was authenticated by the supplier using short tandem repeat (STR) profiling. All cells were cultured at 37°C with 5% $CO_2$ and regularly tested to confirm the absence of mycoplasma contamination.

For transfection, cells were seeded into 96-well plates and transfected at approximately 80% confluence with a total of 225 ng plasmid per well (150 ng SaCas9 or ABE plasmid +75 ng sgRNA plasmid) using JetPRIME (PolyPlus) following the manufacturer's instructions.

Genomic DNA was isolated 72 hours post-transfection using QuickExtract DNA Extraction Solution (Lucigen) following the manufacturer's instructions.

## Deep sequencing and data analysis

Cell lysate were subjected to PCR amplification of the target loci using specific primers (*Supplementary file 2*) with barcodes. PCR was performed using the KAPA HiFi PCR Kit (Roche). Equal amounts of PCR products were pooled, purified, and commercially sequenced (Genewiz) using the NovaSeq platform.

Raw FASTQ reads were demultiplexed using bcl2fastq (Illumina) and analyzed using CRISPResso2 (*Clement et al., 2019*) by aligning amplicon reads to the reference sequences.

## Off-target (OT) effect analysis

Potential OT sites (OTS) were predicted using Cas-OFFinder (*Bae et al., 2014*). The OTS information is provided in *Supplementary file 2*. Predicted OTS regions were amplified by PCR using primers

listed in *Supplementary file 2*, followed by deep sequencing. The resulting reads were analyzed to quantify any OT events involving base modifications.

## Statistical analysis and reproducibility

All experiments included at least three biological replicates. Data are presented as mean ± s.d. Statistical analyses were performed using GraphPad Prism 9. For two-group comparisons, statistical significance was evaluated using a two-tailed unpaired Student's *t*-test.

## Code availability

The UniDesign program and associated scripts, along with detailed usage documentation, are publicly available at https://doi.org/10.5281/zenodo.18058217.

## Materials availability

Materials are available upon request to the corresponding authors.

## Acknowledgements

This work was supported by the National Institutes of Health grants (GM149016 to Xiaoqiang Huang and Xiaofeng Xia; HL164205 to Jie Xu). We thank the Advanced Research Computing (ARC) at the University of Michigan for providing the computational resources and services that supported this research.

## Additional information

### Competing interests

Xiaofeng Xia: is affiliated with ATGC Inc. The other authors declare that no competing interests exist.

### Funding

| Funder | Grant reference number | Author |
| --- | --- | --- |
| National Institutes of Health | GM149016 | Xiaofeng Xia Xiaoqiang Huang |
| National Institutes of Health | HL164205 | Jie Xu |

The funders had no role in study design, data collection and interpretation, or the decision to submit the work for publication.

### Author contributions

Youcai Xiong, Data curation, Formal analysis, Validation, Investigation, Visualization, Methodology, Writing – original draft, Writing – review and editing; Li-Kuang Tsai, Jun Zhou, Shuang Chen, Investigation; Xiaofeng Xia, Supervision, Funding acquisition, Writing – review and editing; Jifeng Zhang, Y Eugene Chen, Supervision, Writing – review and editing; Jie Xu, Resources, Supervision, Funding acquisition, Writing – review and editing; Xiaoqiang Huang, Conceptualization, Resources, Data curation, Software, Formal analysis, Supervision, Funding acquisition, Validation, Investigation, Visualization, Methodology, Writing – original draft, Project administration, Writing – review and editing

### Author ORCIDs

Jifeng Zhang ⓘD https://orcid.org/0000-0001-5161-4705
Xiaoqiang Huang ⓘD https://orcid.org/0000-0002-1005-848X

Reviewer #1 (Public review): https://doi.org/10.7554/eLife.110906.3.sa1
Reviewer #2 (Public review): https://doi.org/10.7554/eLife.110906.3.sa2
Reviewer #3 (Public review): https://doi.org/10.7554/eLife.110906.3.sa3
Author response https://doi.org/10.7554/eLife.110906.3.sa4

## Additional files

### Supplementary files

Supplementary file 1. Spreadsheet containing UniDesign energetic analysis for single, double, and triple mutants.

Supplementary file 2. Spreadsheet containing DNA sequences of constructs, sgRNA sequences, PCR primers, and predicted off-target site sequences used in this study.

MDAR checklist

### Data availability

All computational protein design data, including designed models and summaries of energetic analyses, and the deep sequencing data are publicly available at https://doi.org/10.5281/zenodo.18058217.

The following dataset was generated:

| Author(s) | Year | Dataset title | Dataset URL | Database and Identifier |
|---|---|---|---|---|
| Huang X | 2026 | Fully computational design of SaCas9 with relaxed PAM requirement | https://doi.org/10.5281/zenodo.18058217 | Zenodo, 10.5281/zenodo.18058217 |

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
