## [Editor Report · eLife Assessment]

This **important** study demonstrates the power of the UniDesign computational framework in prospectively engineering a PAM-relaxed *Staphylococcus aureus* Cas9 variant with editing performance comparable to evolution-derived counterparts. The authors responded promptly and thoroughly to reviewer concerns and strengthened the manuscript with additional experimental validation, providing **compelling** evidence through expanded biochemical characterization across multiple human cell types, comprehensive deep-sequencing analyses, and direct comparisons with established variants that illuminate the mechanistic basis of PAM specificity remodeling and Cas9 optimization. By establishing computational design as a rigorous and viable alternative to directed evolution for CRISPR systems, this work will be of broad interest to the protein engineering, genome engineering, synthetic biology, and computational protein design communities.

---

## [Referee Report · Reviewer #1 (Public review)]

[Editors' note: The Reviewing Editor has assessed the work without involving the previous reviewers, updating the eLife Assessment accordingly. The authors did an excellent job of addressing the reviewers' comments and suggestions. The manuscript is now in line with the minor suggestions from the original reviewers, who were already enthusiastic about the first version.]

Summary:

This manuscript by Xiong and colleagues presents a compelling validation of UniDesign, a fully computational protein design framework, by using it to engineer a novel, PAM-relaxed variant of *Staphylococcus aureus* Cas9 (SaCas9) named KRH. The core achievement is the successful de novo generation of a high-performance nuclease (E782K/N968R/R1015H) solely through in silico modeling, without any subsequent experimental optimization or directed evolution. The authors demonstrate that KRH expands the SaCas9 PAM specificity from NNGRRT to NNNRRT, achieving genome editing and base editing efficiencies across multiple human cell types that are comparable to, and sometimes exceed, the well-known evolution-derived KKH variant. The work positions UniDesign not merely as an analytical tool, but as a powerful engine for the generative design of complex molecular functions, offering a scalable and mechanistically insightful alternative to traditional experimental screening.

Strengths:

This is an outstanding manuscript that serves as a powerful proof-of-concept for the next generation of computational protein design. The primary selling point-the raw predictive and generative power of UniDesign-is convincingly demonstrated throughout.

The manuscript shows that the tool can:

(1) successfully navigate a complex sequence landscape to identify a minimal set of three mutations (KRH) that remodel a critical protein-DNA interface;

(2) accurately model and balance the delicate interplay between specific base contacts and non-specific backbone interactions to achieve relaxed PAM specificity;

(3) deliver a final product whose performance is indistinguishable from, and in some cases superior to, a variant that required extensive wet-lab evolution.

The experimental validation is rigorous, thorough, and directly supports the computational predictions. This work will stand as a landmark study for the field, illustrating that computational design has matured to the point where it can reliably generate sophisticated tools for genome engineering.

(1) Demonstration of Generative Power:

The most significant finding is that UniDesign, without any experimental feedback, generated a variant (KRH) that matches the performance of the evolution-derived KKH. This is a remarkable achievement. The iterative design strategy-first reducing PAM bias (R1015H), then restoring binding through non-specific interactions (e.g., N968R, E782K)-is a textbook example of rational design, but it is executed entirely by the algorithm. This validates UniDesign's energy function and search algorithm as capable of capturing the subtle biophysical principles governing PAM recognition.

(2) Mechanistic Insight as a Built-in Feature:

A key advantage of UniDesign highlighted by this work is its inherent ability to provide mechanistic explanations. The computational models not only predicted which mutations would work (e.g., N968R over N968K in the KRH variant) but also why they work. The structural and energetic analyses showing the bidentate salt bridge formed by Arg968 versus the single bond formed by Lys968 (Figure 4A) is a perfect example of how the tool's output can rationalize functional differences, a level of insight that is rarely attainable from directed evolution campaigns alone.

(3) Scalability and Accessibility for Engineering:

The authors explicitly contrast UniDesign's efficiency (minutes to hours per design run) with the computational expense of methods like COMET and the experimental overhead of directed evolution. The improvements to UniDesign v1.2, specifically the mutation-count and sequence-uniqueness penalties, directly address a key challenge in computational design (generating diverse, low-energy point-mutant libraries). This positions the tool as a highly accessible and scalable platform for engineering other CRISPR systems, a point that will be of immense interest to the community.

---

## [Referee Report · Reviewer #2 (Public review)]

Summary:

This manuscript describes the fully in silico design of a new variant of *Staphylococcus aureus* Cas9 (SaCas9) using an improved UniDesign workflow.

The design strategy consists of three sequential steps:

(1) Reducing positional bias at PAM position 3;

(2) Restoring DNA binding through nonspecific interactions;

(3) Combining individually favorable substitutions.

The overall pipeline is conceptually elegant and logically structured, and the genome-editing activity of the designed variants is comprehensively characterized. The resulting KRH variant exhibits relaxed PAM specificity, expanding the targeting range of SaCas9 across diverse cell types. Notably, the KRH variant demonstrates performance comparable to that of the evolution-derived KKH variant, underscoring the effectiveness of the proposed computational design framework.

---

## [Referee Report · Reviewer #3 (Public review)]

Summary:

This study reports KRH, a SaCas9 variant computationally engineered via UniDesign to recognize an expanded NNNRRT PAM with substantially enhanced editing efficiency at non-canonical sites. KRH achieves genome- and base-editing efficiencies comparable to or exceeding the evolution-derived KKH variant across multiple human cell types, demonstrating that computational design can effectively remodel PAM specificity while preserving nuclease activity.

Strengths:

The research follows a clear line of reasoning, and the results appear sound. The computational design strategy presented offers a valuable alternative to directed evolution, with potential applicability beyond Cas9 engineering.

---

## [Author Response]

The following is the authors’ response to the original reviews.

**Public Reviews:**

**Reviewer #1 (Public review):**
Summary:This manuscript by Xiong and colleagues presents a compelling validation of UniDesign, a fully computational protein design framework, by using it to engineer a novel, PAM-relaxed variant of *Staphylococcus aureus* Cas9 (SaCas9) named KRH. The core achievement is the successful de novo generation of a high-performance nuclease (E782K/N968R/R1015H) solely through in silico modeling, without any subsequent experimental optimization or directed evolution. The authors demonstrate that KRH expands the SaCas9 PAM specificity from NNGRRT to NNNRRT, achieving genome editing and base editing efficiencies across multiple human cell types that are comparable to, and sometimes exceed, the well-known evolution-derived KKH variant. The work positions UniDesign not merely as an analytical tool, but as a powerful engine for the generative design of complex molecular functions, offering a scalable and mechanistically insightful alternative to traditional experimental screening.Strengths:This is an outstanding manuscript that serves as a powerful proof-of-concept for the next generation of computational protein design. The primary selling point-the raw predictive and generative power of UniDesign-is convincingly demonstrated throughout.The manuscript shows that the tool can:(1) successfully navigate a complex sequence landscape to identify a minimal set of three mutations (KRH) that remodel a critical protein-DNA interface;(2) accurately model and balance the delicate interplay between specific base contacts and non-specific backbone interactions to achieve relaxed PAM specificity;(3) deliver a final product whose performance is indistinguishable from, and in some cases superior to, a variant that required extensive wet-lab evolution.The experimental validation is rigorous, thorough, and directly supports the computational predictions. This work will stand as a landmark study for the field, illustrating that computational design has matured to the point where it can reliably generate sophisticated tools for genome engineering.(1) Demonstration of Generative Power:The most significant finding is that UniDesign, without any experimental feedback, generated a variant (KRH) that matches the performance of the evolution-derived KKH. This is a remarkable achievement. The iterative design strategy-first reducing PAM bias (R1015H), then restoring binding through non-specific interactions (e.g., N968R, E782K)-is a textbook example of rational design, but it is executed entirely by the algorithm. This validates UniDesign's energy function and search algorithm as capable of capturing the subtle biophysical principles governing PAM recognition.(2) Mechanistic Insight as a Built-in Feature:A key advantage of UniDesign highlighted by this work is its inherent ability to provide mechanistic explanations. The computational models not only predicted which mutations would work (e.g., N968R over N968K in the KRH variant) but also why they work. The structural and energetic analyses showing the bidentate salt bridge formed by Arg968 versus the single bond formed by Lys968 (Figure 4A) is a perfect example of how the tool's output can rationalize functional differences, a level of insight that is rarely attainable from directed evolution campaigns alone.(3) Scalability and Accessibility for Engineering:The authors explicitly contrast UniDesign's efficiency (minutes to hours per design run) with the computational expense of methods like COMET and the experimental overhead of directed evolution. The improvements to UniDesign v1.2, specifically the mutation-count and sequence-uniqueness penalties, directly address a key challenge in computational design (generating diverse, low-energy point-mutant libraries). This positions the tool as a highly accessible and scalable platform for engineering other CRISPR systems, a point that will be of immense interest to the community.

We sincerely thank the reviewer for the comprehensive summary and the highly positive and encouraging comments on our manuscript.

Weaknesses:(1) Title and Abstract Emphasis:The title and abstract are effective but could be slightly sharpened to emphasize the primary message. Consider a title like "Fully computational design of a PAM-relaxed SaCas9 variant with UniDesign demonstrates power to match directed evolution." The abstract could more explicitly state upfront that the design was achieved without any experimental iteration.

Thank you for this valuable suggestion. We have revised the title and abstract accordingly to better reflect your feedback.

(2) Figure 1, Panel M:The data points in panel M are currently presented at a font size that makes them difficult to read, particularly the labels for the many triple-mutant variants. This density obscures the clear identification of the top-performing designs, such as the KRH variant selected for experimental validation. I recommend that the authors increase the font size of all text elements within this panel, including axis labels, tick marks, and data point labels, to improve legibility. If necessary, the panel dimensions can be adjusted or the layout reorganized to accommodate the larger text without compromising clarity. Ensuring this figure is readable is important, as it visually communicates the energetic convergence that led to the selection of KRH.

Thank you for this helpful suggestion. We have increased the font size the Figure 1M, as well as in Figure 1C and Figure 1E, to improve the readability in the revised manuscript.

(3) Generality of the Design Strategy for Other PAM Positions:The design strategy focused on relaxing specificity at the highly constrained third position of the PAM (the guanine in NNGRRT). How transferable is this specific strategy (i.e., disrupting a key specific contact and compensating with non-specific backbone binders) to relaxing other positions in the PAM or to other Cas enzymes with different PAM-interaction architectures? A short discussion on this point would help readers understand the broader applicability of the "fine-tuning the balance" principle.

Thank you for this insightful question and suggestion. The current study builds upon our previous work on CRISPR–Cas PAM recognition modeling using UniDesign (PMID: 37078688), in which eight Cas9 proteins and two Cas12 proteins (each has a different PAM) were investigated. Our computational results demonstrated that UniDesign can effectively capture the mutual preferences between natural PAMs and native PAM-interacting amino acids (PIAAs). For example, UniDesign accurately predicted the canonical PAMs of SpCas9 and SaCas9 as NGG and NNGRRT, respectively; conversely, given their canonical PAMs, UniDesign successfully recapitulated the corresponding PIAAs in both systems.

These findings provide the foundation for the present study and motivate our selection of SaCas9 as a representative system to explore PAM relaxation, thereby further demonstrating UniDesign’s predictive power through experimental validation. Although we did not perform similar PAM relaxation designs for other Cas9 or Cas12 proteins, we believe that the UniDesign framework is broadly generalizable and can be readily extended to these systems. We have included additional discussion to clarify this point and highlight the broader applicability of our design strategy.

**Reviewer #2 (Public review):**
Summary:This manuscript describes the fully in silico design of a new variant of *Staphylococcus aureus* Cas9 (SaCas9) using an improved UniDesign workflow.The design strategy consists of three sequential steps:(1) reducing positional bias at PAM position 3;(2) restoring DNA binding through nonspecific interactions;(3) combining individually favorable substitutions.The overall pipeline is conceptually elegant and logically structured, and the genome-editing activity of the designed variants is comprehensively characterized. The resulting KRH variant exhibits relaxed PAM specificity, expanding the targeting range of SaCas9 across diverse cell types. Notably, the KRH variant demonstrates performance comparable to that of the evolution-derived KKH variant, underscoring the effectiveness of the proposed computational design framework.Strengths:The design pipeline is entirely computational and does not rely on experimental data for pretraining or iterative optimization.

We thank the reviewer for the concise and accurate summary of our manuscript.

Weaknesses:The computationally generated KRH mutant differs from the experimentally evolved KKH variant by only a single residue, which may reflect insufficient exploration of the available sequence space.

Thank you for this insightful critique. In the present study, our strategy was not to allow UniDesign to freely explore all 27 mutable positions simultaneously, but rather to constrain the search to point mutations (e.g., double or triple mutants) within the full sequence space (approximately 20^27^). Even with this constraint, UniDesign effectively samples a substantially large design space compared to traditional protein engineering approaches.

Through iterative design, we observed that only certain residue types became enriched at a subset of positions when identifying effective double mutants. These enriched residues were then systematically combined to generate performance-enhancing triple mutants in an automated manner. Although we ultimately selected the KRH mutant for experimental validation due to its high similarity to the known KKH variant, UniDesign also proposed additional multi-mutants that are distinct from KKH (see Figure 1M).

**Reviewer #3 (Public review):**
Summary:This study reports KRH, a SaCas9 variant computationally engineered via UniDesign to recognize an expanded NNNRRT PAM with substantially enhanced editing efficiency at non-canonical sites. KRH achieves genome- and base-editing efficiencies comparable to or exceeding the evolution-derived KKH variant across multiple human cell types, demonstrating that computational design can effectively remodel PAM specificity while preserving nuclease activity.Strengths:The research follows a clear line of reasoning, and the results appear sound. The computational design strategy presented offers a valuable alternative to directed evolution, with potential applicability beyond Cas9 engineering.

We thank the reviewer for the concise and accurate summary of our manuscript.

Weaknesses:The benchmarking of the UniDesign method is insufficient. How its performance compares to other protein design algorithms, whether the energy function parameters were systematically optimized, and if the design strategy can be generalized to other Cas9 orthologs or genome engineering tasks.

Thank you for this valuable critique. The present study builds upon our previous work on CRISPR–Cas PAM recognition modeling using UniDesign (PMID: 37078688), in which many of these concerns were systematically addressed. In that study, UniDesign was benchmarked against Rosetta, a well-established protein design platform, across eight Cas9 proteins and two Cas12 proteins, each recognizing distinct PAM sequences.

Our results demonstrated that UniDesign effectively captures the mutual preferences between natural PAMs and native PAM-interacting amino acids (PIAAs) across these CRISPR–Cas systems. For example, UniDesign accurately predicted the canonical PAMs of SpCas9 and SaCas9 as NGG and NNGRRT, respectively; conversely, given their canonical PAMs, UniDesign successfully recapitulated the corresponding PIAAs in both systems.

These findings provide the foundation for the present study and motivate our selection of SaCas9 as a representative system to explore PAM relaxation, thereby further demonstrating UniDesign’s predictive power through experimental validation. Although we did not perform analogous PAM relaxation designs for other Cas9 or Cas12 proteins in this work, we believe that the UniDesign framework is broadly generalizable and can be readily extended to these systems. We have incorporated additional discussion in the revised manuscript to address these points and clarify the broader applicability of our approach.

**Recommendations for the authors:**

**Reviewer #2 (Recommendations for the authors):**
(1) SaCas9 is highlighted for its AAV compatibility, but the manuscript does not further discuss how the KRH variant may benefit AAV-based genome editing applications. A brief discussion on how expanded PAM compatibility could facilitate target selection in AAV-constrained therapeutic settings would strengthen the translational relevance of the work, potentially reducing the need for split-Cas9 or dual-vector strategies.

Thank you for your helpful suggestion. We have added a brief discussion in the revised manuscript highlighting how the KRH variant’s expanded PAM compatibility may enhance AAV-based genome editing applications. Specifically, this property can broaden the range of targetable genomic sites and may reduce the need for split-Cas9 or dual-vector delivery strategies in size-constrained AAV therapeutic contexts.

(2) The study shows that a fully computational workflow can recapitulate the performance of an evolution-derived variant. A short discussion comparing the scalability and practical advantages of computational design versus directed evolution for future PAM engineering would help emphasize the broader methodological significance of UniDesign.

Thank you for your valuable suggestion. We have added a brief discussion in the revised manuscript comparing the scalability and practical advantages of computational design with directed evolution for PAM engineering. Specifically, we highlight that UniDesign enables rapid and scalable exploration of sequence space without requiring iterative experimental screening, thereby offering a complementary—and in some cases more efficient—approach to directed evolution for future protein engineering applications.

(3) The noticeable variation in editing efficiency across cell types, particularly the lower activity in A549 cells. Could the authors explain why the differences in editing efficiency are so large?

Thank you for this insightful comment. We agree that the variation in editing efficiency across cell types—particularly the lower activity observed in A549 cells—warrants clarification, and we have added a corresponding discussion in the revised manuscript. We attribute this observation to two main factors. First, transfection efficiency varies substantially across cell lines; in our experiments, A549 cells exhibited lower transfection efficiency compared to HEK293T, HeLa, and U2OS cells, which likely contributes to the reduced editing efficiency. Second, the intrinsic performance of genome editing systems can differ across cellular contexts due to variations in DNA repair pathways, including chromatin accessibility and the expression levels of key repair-related genes. Importantly, despite this cell-type-dependent variability in absolute editing efficiency, the KRH variant consistently outperformed wild-type SaCas9 across all tested cell lines, underscoring the robustness and general applicability of our design.

(4) Given that the computationally generated KRH mutant differs from the experimentally evolved KKH variant by only a single residue, it would be valuable to discuss whether R968 (or saturation mutations at this site) has previously been explored experimentally, and to elaborate on strategies for further expanding the diversity of mutations identified through the computational design framework.

Thank you for your suggestion. We have added a brief discussion in the manuscript noting that, to the best of our knowledge, R968 has not been experimentally characterized prior to this study. It was identified solely through our computational design workflow, highlighting the strength of our approach.

**Reviewer #3 (Recommendations for the authors):**
(1) During the protein amino acid conformational sampling process in UniDesign, were nucleic acid conformational changes taken into consideration?

Thank you for this question. Nucleic acid conformational changes were not explicitly considered during the protein sequence design stage in UniDesign after the four specific PAM variants (e.g., TTAGGT, TTCGGT, TTGGGT, and TTTGGT) were defined. We consider this assumption reasonable, as the base conformations in these PAM sequences are expected to remain largely stable, with minimal structural variation due to preserved base-stacking interactions.

(2) The authors used a mutation-count penalty to control the number of mutations generated during the design process, which appears to occasionally yield results that exceed the intended limit. Is this an efficient approach? Could the count be controlled more directly by imposing constraints within the design procedure itself?

Thank you for these insightful questions. You are correct that the design process may occasionally yield variants exceeding the intended mutation limit. This occurs because the mutation-count penalty is implemented as a soft constraint, where violations incur a penalty rather than being strictly excluded. Based on our benchmarking, this strategy—combined with the duplicate-design penalty—has been effective in generating multimutant variants with mutation counts close to the desired range. However, we acknowledge that this approach may not achieve optimal efficiency. We are currently developing improved strategies in UniDesign to more directly control mutation counts by incorporating explicit constraints during the sequence simulation process, which we expect will further enhance design precision and efficiency.

(3) Is the new version of UniDesign developed specifically for the Cas9 design task in this study? What are its advantages and disadvantages compared to other state-of-the-art protein design algorithms?

Thank you for this important question. The new version of UniDesign (v1.2) was not developed specifically for Cas9 engineering. Rather, it is intended as a general framework for protein engineering tasks that focus on introducing point mutations to improve protein properties, as opposed to de novo design. Compared to current state-of-the-art protein design methods—many of which are deep learning–based—UniDesign offers distinct advantages and limitations. Deep learning approaches are often highly efficient and powerful but may lack interpretability in their predictions. In contrast, UniDesign is a well-benchmarked, lightweight, physics-based method that provides greater interpretability, allowing users to better understand the underlying basis of the design decisions. On the other hand, a limitation of UniDesign is that it is less straightforward to incorporate experimental feedback for iterative refinement, such as fine-tuning the scoring function for specific design tasks.

(4) The study employed a three-round design process to obtain the mutants. Is there a conformational correlation between the mutation sites identified in these three rounds? Could this have been accomplished in a single computational run instead of three separate calculations?

Thank you for these insightful questions. We adopted a multi-round design strategy for SaCas9 PAM relaxation because this task inherently involves multi-objective optimization: enhancing PAM compatibility—particularly relaxing base recognition at the third PAM position—while preserving editing activity comparable to wild-type SaCas9. In our view, identifying the key mutations (e.g., E782K, N968R, and R1015H) in a single UniDesign run would be highly challenging due to competing energetic requirements. In the first round, R1015H emerged from single-site mutational scanning as the most favorable PAM-relaxing mutation based on its minimal MAD score. However, this mutation also significantly increased the binding energy relative to wild-type SaCas9 with its native PAM, suggesting a likely reduction in editing activity due to weakened binding. To address this, the second round focused on compensatory mutations. Variants such as E782K and N968R (along with several additional candidates) were identified in the context of R1015H to reduce binding energy and partially restore affinity. In the third round, we further combined compatible mutations from the second round, resulting in variants that more effectively lowered binding energy and restored it to levels comparable to wild-type SaCas9 with its native PAM. Notably, the design objectives in rounds one and two drive binding energy in opposite directions, making it unlikely that all key mutations could be identified simultaneously in a single run. During the design process, we also observed conformational correlations among mutation sites. For example, R1015H can form hydrogen-bonding interactions with residue E993, and we observed multiple alternative mutations at position 993 (e.g., E993S, E993P, E993A, E993G, E993K, and E993R), suggesting local structural coupling between these positions.

(5) In Figure 4D, for the FANCF-1 site, there appears to be a noticeable difference in editing efficiency between KKH-ABE and KRH-ABE. Is this difference statistically significant? If so, please provide an explanation for this observation.

Thank you for this question. For the FANCF-1 site shown in Figure 4D, we performed statistical analyses and found that the differences in editing efficiency between KKH-ABE and KRH-ABE are not statistically significant: *P*(A4) = 0.1239, *P*(A10) = 0.0671, *P*(A12) = 0.0942, and *P*(A13) = 0.1349 (two-tailed unpaired Student’s *t*-test). These results indicate that KRH-ABE and KKH-ABE exhibit comparable editing efficiencies at this site, supporting our overall conclusion that the computationally designed KRH variant achieves performance on par with the KKH variant.

(6) Does the evolutionary term within the UniDesign scoring function bias the designed sequences towards pre-existing protein features?

Thank you for this question. In this study, as well as in our previous work on Cas9 PAM recognition modeling (PMID: 37078688), the evolutionary term in the UniDesign scoring function was completely disabled. Therefore, it does not introduce any bias toward pre-existing protein features in the designed sequences.